# Multifunctional Cu-Se Alloy Core Fibers and Micro–Nano Tapers

**DOI:** 10.3390/nano13040773

**Published:** 2023-02-19

**Authors:** Min Sun, Yu Liu, Dongdan Chen, Qi Qian

**Affiliations:** State Key Laboratory of Luminescent Materials and Devices, Institute of Optical Communication Materials, Guangdong Provincial Key Laboratory of Fiber Laser Materials and Applied Techniques, and Guangdong Engineering Technology Research and Development Center of Special Optical Fiber Materials and Devices, School of Materials Science and Engineering, South China University of Technology, Guangzhou 510640, China

**Keywords:** Cu-Se alloy, multifunctional fiber, thermal drawing, photoelectric detection, thermoelectric conversion

## Abstract

Cu-Se alloy core fibers with glass cladding were fabricated by a thermal drawing method of a reactive molten core. The composition, crystallography, and photoelectric/thermoelectric performance of the fiber cores were investigated. The X-ray diffraction spectra of the Cu-Se alloy core fibers illustrate the fiber cores being polycrystalline with CuSe and Cu_3_Se_2_ phases. Interestingly, the fiber cores show a lower electrical conductivity under laser irradiation than under darkness at room temperature. Meanwhile, the fiber cores possess a power factor of ~1.2 mWm^−1^K^−2^ at room temperature, which is approaching the value of the high thermoelectric performance bulk of Cu_2_Se polycrystals. The flexible Cu-Se fibers and their micro–nano tapers have potential multifunctional applications in the field of photoelectric detection and thermoelectric conversion on curved surfaces.

## 1. Introduction

Copper selenide (CuSe) is a semiconductor material with excellent electrical and optical properties [1]. It is available in many crystalline phases and structures: the stoichiometric compounds of Cu_2_Se, Cu_3_Se_2_, CuSe, or CuSe_2_, and the non-stoichiometric Cu_2-x_Se. Two-dimensional thin films of CuSe have been used in many applications, such as solar cells, photodetectors, thermoelectric devices, and gas sensors [2,3,4,5,6,7]. CuSe is reported to be hexagonal at room temperature, transforming to orthorhombic at 321 K, and returning to hexagonal at 393 K [8]. For Cu_1.8_Se, the direct band gap width is 2.2 eV, and for Cu_2_Se, the indirect band gap width is 1.4 eV [9]. However, the forbidden bandwidth of CuSe is 1.2 eV. There are various reasons for the wide variation of the band gap, including large stoichiometric deviations, a large number of dislocations, and quantum size effects [10]. In addition to studying the crystalline phases of CuSe, researchers have done many studies on the morphologies and electronic bands of CuSe compounds, including nanoparticles, nanotubes, and nanowires [11,12,13,14,15]. However, the reported CuSe materials only show sole-function applications and the multifunctional performance of CuSe materials should be systematically assessed.

In the field of photodetection, Kou et al. successfully prepared In^3+^-doped Cu_2-x_Se nanostructures using an electrochemical deposition method [16]. It was analyzed that the doping of In^3+^ caused a larger photo-contact surface in the nanostructure, which facilitated the rapid separation of photogenerated charges, thus increasing the photocurrent. The results show that In^3+^ doping can improve the photoelectric properties of Cu_2-x_Se and has potential applications for photodetection devices. Furthermore, one of the extensive research fields of Cu-Se compounds is thermoelectric conversion. In 2013, Liu et al. investigated the phase transition properties of Cu_2-x_Se, which led to improved thermal properties, and electrical properties by electron and phonon critical scattering [17]. In Cu_2-x_Se, Se forms a face-centered cubic sublattice, which provides a pathway for holes in the semiconductor. Cu ions are highly disordered around the Se sublattice and liquid-like in their mobility, resulting in Cu_2-x_Se having a very low thermal conductivity. It resulted in Cu_2-x_Se with ultra-high thermoelectric properties, and the highest *ZT* value of 2.3 at room temperature. Cu_2-x_Se has been expected to be used for microprocessor cooling and thermoelectric generators to power wireless sensors.

Herein, Cu-Se alloy core fibers were prepared with a thermal drawing method with a reactive molten core. The fiber cores showed a photoelectric response with a lower electrical conductivity under laser irradiation than under darkness. The fiber cores possessed high electrical conductivities and high Seebeck coefficients, approaching that of the bulk Cu_2-x_Se polycrystals [17]. The Cu-Se alloy core fibers with mechanical flexibility can be applied in flexible photoelectric detection, optical switching, thermoelectric sensing, and even multifunctional fiber sensing.

## 2. Experimental Procedure

The precursor powders of the fiber core were chosen to be CuSe. The CuSe used in this experiment was purchased from Longjin Materials Corporation, Shanghai, and it has a melting point of about 660 K and a density of about 6.8 g/cm^3^. The CuSe powders were stacked into a BK7 glass tube purchased from Schott Corporation, Germany, and it possesses about a 1070 K softening temperature. Cu-Se core fibers with several meters in length were thermally drawn at about 1150 K in an optical fiber tower under argon. Additionally, some micro–nano Cu-Se fiber tapers were secondly drawn from the 200-μm-diameter Cu-Se fibers with an alcohol lamp.

The morphology and elemental distribution on the fiber end face were analyzed by an electron probe X-ray microanalysis (Shimadzu EPMA-1600). The crystallography of the cores was characterized by X-ray diffraction (XRD, X’Pert Pro) and microscopic Raman spectroscopy (Renishaw RM2000). The fiber cores were ground and analyzed by a UV-NIR spectrometer (Perkin-Elmer Lambda). The photocurrents of the fibers under 532/808 nm laser irradiation were measured by connecting both ends of the fiber to an external circuit [18]. Additionally, the Seebeck coefficient and electrical conductivity were measured by a two-probe method, when two ends of the fibers or tapers are silver-pasted and connected to the electrical circuit [19]. Under the same conditions, three-time measurements were performed to obtain the average measuring values, and their relative deviations were smaller than 5%.

## 3. Results and Discussion

Figure 1a shows the cross-section electron micrograph of the Gu-Se alloy core fiber. It can be seen that the core diameter is about 380 μm and the core/cladding structure is intact, indicating that the borosilicate glass and Cu-Se core have high-temperature wettability [20]. Figure 1b–e show the element distribution of the wavelength dispersive spectrometer (WDS) on the fiber end face. The boundary of the element distribution forms a circle, while Cu and Se are mainly distributed in the core region, and Si and O are mainly distributed in the cladding region. By the WDS of the electron probe micro analyzer (EPMA), the atomic ratio of Cu and Se was determined to be Cu_1.2_Se. It is worth noting that both Figure 1b,c exhibit elemental enrichment, and the distribution of elements is not uniform. This implies that during the high-temperature drawing process, CuSe underwent some kind of Se element volatilization, and the atomic ratio of Cu and Se changed.

Figure 2a shows the XRD spectra of the fiber cores after drawing. It can be seen that the XRD peaks indicate the crystalline phases of Cu_3_Se_2_ (JCPDS#47-1745) and CuSe (JCPDS#34-0171). Figure 2b shows the Raman spectra of the precursor CuSe powders and Cu-Se fiber cores. In the range of 200–300 cm^−1^, there is a peak at 262 cm^−1^ in the CuSe powders, which corresponds to the vibration of Cu-Se. The peak at 262 cm^−1^ is also present in the spectrum of the Cu-Se powders, and a new peak at 193.0 cm^−1^ is present. This peak position is consistent with that reported in the literature for Cu_3_Se_2_ [21], indicating that the Cu_3_Se_2_ crystalline phase was generated during the fiber drawing process, and that the final fiber crystalline phase composition was a mixture of CuSe and Cu_3_Se_2_.

The reflectance spectra of the CuSe core precursor powder and Cu-Se core composite glass fiber core can be obtained via a UV-NIR spectrometer, as shown in Figure 3. The CuSe raw powders and Cu-Se core powders have a strong absorption, and the strongest absorption is in the 500~1000 nm wavelength range. Therefore, visible light was chosen as the light source for the subsequent photocurrent test. The transmittance of Cu-Se core powders in this region is higher than that of the CuSe raw powders. The reason is that Cu_3_Se_2_ is generated during the CuSe fiber drawing process. The forbidden bandwidth is E_g_ = 2.03 eV for CuSe [22] and E_g_ = 1.45 eV for Cu_3_Se_2_ [23], due to the intrinsic absorption within the semiconductor material of sufficient energy for the photons to be absorbed. A photon of sufficient energy excites an electron, which leaps across the forbidden band into the empty conduction band and leaves a hole in the valence band. Due to the narrowing of the forbidden band width, photons can cross the forbidden band more easily, and the Cu-Se alloy core fiber can exhibit a stronger absorption of light. The photocurrents of the Cu-Se alloy core fiber in dark and irradiation conditions are carried out using the same experimental setup as our previous study [18], which was mainly focused on the thermoelectric and mechanical performance of Bi_2_Te_3_ micro–nano fibers without any optical irradiation. Figure 3b shows the current–voltage curves of the Cu-Se core fiber in dark and irradiation conditions. It can be seen that the electrical conductivity of the Cu-Se core fiber becomes significantly smaller under the irradiation of 532 nm laser and 808 nm laser. By linear fitting and calculation, the conductivity of the Cu-Se core fiber was reduced by 60% under the 532 nm laser and 80% under the 808 nm laser. According to the reported literature [22,23], the response of Cu_3_Se_2_ to light irradiation is mainly in the form of conductivity reduction.

Beyond these, the Cu-Se alloy core fiber was secondly drawn by an alcohol lamp to be a micro–nano taper with a maximum diameter of about 200 μm and a minimum diameter lower than 1 μm. As shown in Figure 4a, two ends of a fiber or a taper were silver-pasted and connected to the electrical circuit. Additionally, when one end was heated at the groove and the other end was cooled at the sink, the thermoelectric voltage differences arose with temperature differences. In Figure 4b, the Seebeck coefficients of the Cu-Se fiber and its micro–nano taper were measured, being 101 μV/K and 117 μV/K, respectively, at room temperature. When the Cu-Se taper possesses a 16% higher Seebeck coefficient than the Cu-Se fiber does, it should be derived from a size-related effect. So the power factors (*PF = S*^2^*σ*) of the fiber and taper were calculated, respectively, being about 1.2 mWm^−1^K^−2^ and 1.6 mWm^−1^K^−2^, approaching that of the bulk Cu_2_Se polycrystals, when the electrical conductivity of the taper was estimated to be 1176 S/cm, the same as the Cu_-_Se fiber cores. Additionally, the *ZT* values of the fibers and tapers were calculated, respectively, being approximately 0.45 and 0.60 at 300 K, when their thermal conductivity was estimated to be about 0.8 Wm^−1^K^−1^, as is reported for the bulk Cu_2_Se polycrystals [17]. The *ZT* value of the fibers at 300 K was similar to that of the bulk Cu_2_Se polycrystals [17], but much lower than the *ZT* value of 2.3 at their phase change point of 400 K. In addition, for estimating their mechanical flexibility [18], the Cu-Se fibers with a diameter (*D*) of 200 μm exhibited a minimum bending radius (*r*_min_) of approximately 2 cm during the bending tests, and a maximum bending strain (*ε*_max_ = *D*/2*r*_min_) of 0.5%. Therefore, Cu-Se alloy core fibers can be expected to be used as multifunctional fibers in the fields of photoelectric detection and thermoelectric conversion on curved surfaces.

## 4. Conclusions

In conclusion, Cu-Se alloy core fibers have been fabricated by using molten-core thermal drawing. The polycrystalline Cu-Se cores are constituted of CuSe and Cu_3_Se_2_, and their composition is made up of Cu_1.2_Se. Interestingly, the electrical conductivity of the Cu-Se cores under laser irradiation is only a third of that under darkness. The as-drawn Cu-Se core fibers possess a *PF* of ~1.2 mWm^−1^K^−2^ and *ZT* of ~0.45. The secondly drawn Cu-Se core tapers possess a *PF* of ~1.6 mWm^−1^K^−2^ and *ZT* of ~0.60. Ultimately, the Cu-Se core shows a good flexibility and photo-/thermo-electric responses at room temperature, and future work should focus on the related size effect, phase change, and optimized applications of these multifunctional fibers.

## Figures and Tables

**Figure 1 nanomaterials-13-00773-f001:**
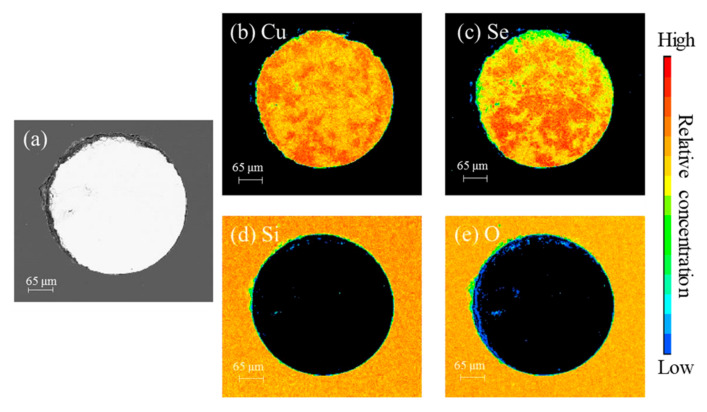
(**a**) Cross-section electronic micrograph of the Cu-Se core fiber; element distribution of (**b**) Cu, (**c**) Se, (**d**) Si, and (**e**) O in the fiber core region. The color palette at the right exhibits the relative concentration of the related element in (**b**–**e**) from low to high.

**Figure 2 nanomaterials-13-00773-f002:**
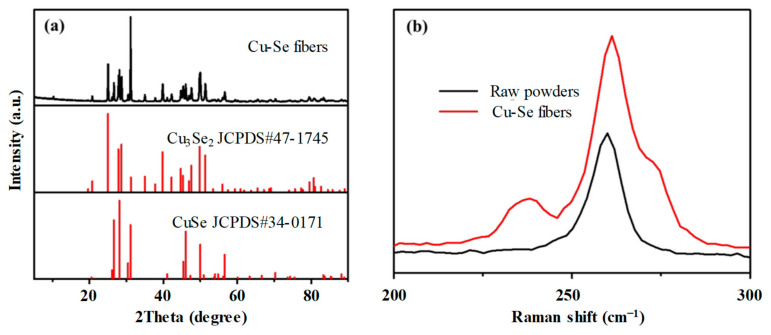
(**a**) X-ray diffraction spectra of the Gu-Se core fiber; and (**b**) Raman spectra of the Gu-Se raw powders and core.

**Figure 3 nanomaterials-13-00773-f003:**
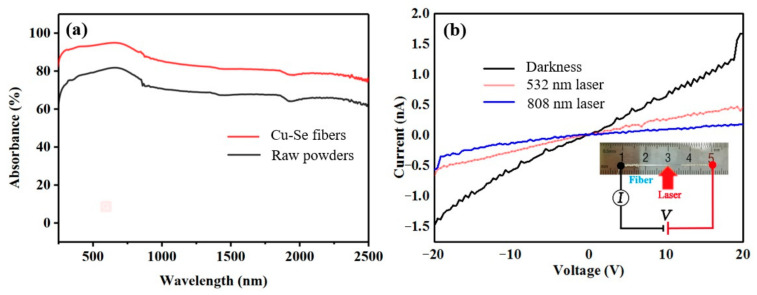
(**a**) Absorbance spectra for CuSe raw powders and Cu-Se core fibers; and (**b**) the current-voltage curves of a Cu-Se core fiber in dark and irradiated conditions. Inset of (**b**) is the optical graph of a Cu-Se fiber and schematic of the electrical circuit.

**Figure 4 nanomaterials-13-00773-f004:**
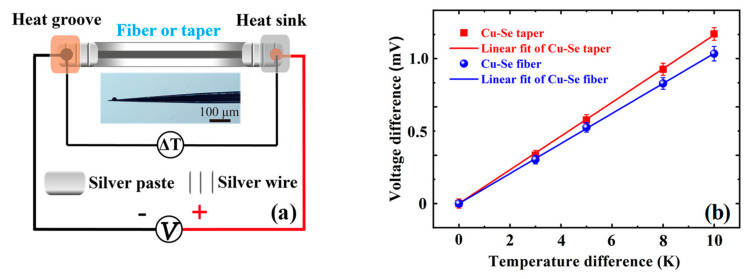
(**a**) Schematic of an electrical circuit connecting the two ends of a fiber or a micro–nano taper; the inset of (**a**) is the optical micrograph of a Cu-Se taper. (**b**) The thermoelectric voltages at temperature differences of 0 K, 3 K, 5 K, 8 K, or 10 K cross the two ends. The error bar in (**b**) shows 5% measurement uncertainty.

## Data Availability

The production data are available on request from the corresponding author.

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
