# Peer review of "Multifunctional Cu-Se Alloy Core Fibers and Micro–Nano Tapers"

_nanomaterials, 2023, doi:10.3390/nano13040773_

Round 1

Reviewer 1 Report

Comments on:

  Multifunctional Cu-Se alloy core fibers

by Min Sun, Yu Liu, Dongdan Chen and Qi Qian

 The paper presents quite simple characterisation study of the Cu-Se alloy core fibres. The results are not particularly interesting but they might have some limited value to a few experts in the field. There is a potential to boost the scientific significance of this publication with a relatively low extra performing a more detailed study (perhaps this is already done).

Nevertheless, the paper is well written and easy to comprehend. I have few comments that are easy to implement and should help a potential reader to get through the material

Figures:

Most figures are not aligned with the text. Please, fix that.

 References:

In each citation there is a symbol “[J]”. The meaning of that is not clear to me.

 Detailed comments:

L11:  “diffraction illustrates the fiber cores being polycrystalline” : please clarify, it is quite confusing now

L26: “photothermal conversion” : conversion is a phenomenon, not an application. Please, phrase.

L29: “x=0.2” : what is x?

L34-36: please, reformulate. The message is not clear

L49: “ZT value of 2.3” : Please, explain and compare to the materials.

L55: “high temperature performance”: it is very vague. Please, be more specific

L56: ”would be applied“: the message of this sentence is not clear.

Figure 1: explain the  colour code.

(b-e) Element distribution of Cu, Se, O, and Si in the fiber core region. -> Element distribution of (b) Cu, (c) Se, (d) O, and (e) Si in the fiber core region.

L83: what is “wave--- spectrometer”?

L86: What is WDS and EPMA? If these are devices, please, rephrase the sentence to make it correct. Right now it is non-sense.

L92: Expand XRD

Figure 3: labels (a) and (b) are missing

Author Response

Responses to the Reviewers’ comments:

Reviewer #1:

Comments on:

The paper presents quite simple characterisation study of the Cu-Se alloy core fibres. The results are not particularly interesting but they might have some limited value to a few experts in the field. There is a potential to boost the scientific significance of this publication with a relatively low extra performing a more detailed study (perhaps this is already done).

Nevertheless, the paper is well written and easy to comprehend. I have few comments that are easy to implement and should help a potential reader to get through the material

Response: We sincerely thank the reviewer for the comment that the manuscript is well-written and easy to comprehend. And the related responses to the comments are attached below. For following the reviewer’s advice and boosting the scientific significance of this manuscript, we have added the size-related effects on the fiber properties in Figure 4 below of the revised manuscript.

Figure 4. (a) Schematic of an electrical circuit connecting the two ends of a fiber or a micro-nano taper; Inset of (a) is the optical micrograph of a Cu-Se taper. (b) The thermoelectric voltages at temperature differences of 0 K, 3 K, 5 K, 8 K, or 10 K cross the two ends. The error bar in (b) shows 5% measurement uncertainty.

 Figures: Most figures are not aligned with the text. Please, fix that.

References: In each citation there is a symbol “[J]”. The meaning of that is not clear to me.

Response: We thank the reviewer for the comment to improve the manuscript. The figure-related revision has been made and all the“[J]” meaning “journal” in the citations have been deleted.

 Detailed comments:

L11: “diffraction illustrates the fiber cores being polycrystalline” : please clarify, it is quite confusing now

Response: “X-ray diffraction illustrates the fiber cores being polycrystalline” has been changed to “X-ray diffraction spectra of the Cu-Se alloy core fibers illustrates the fiber cores being polycrystalline”.

 L26: “photothermal conversion” : conversion is a phenomenon, not an application. Please, phrase.

Response: “thermoelectric/photothermal conversion” has been changed to “thermoelectric devices, and gas sensors”.

 L29: “x=0.2” : what is x?

Response: “Cu2-xSe” has been changed to “Cu1.8Se”.

 L34-36: please, reformulate. The message is not clear

Response: The sentence has been reformulated to “But the reported CuSe materials show sole-function applications and the multifunctional performance of CuSe materials should be systematically assessed.”.

 L49: “ZT value of 2.3” : Please, explain and compare to the materials.

Response: The 0.45-ZT value of the fiber at 300 K is similar to that of the bulk Cu2Se polycrystals [Ref.17 Advanced Materials, 2013, 25: 6607], but much lower than the 2.3-ZT value at their phase change point of ~400 K. And the related revision has been made clear at Line 160-162 of Page 5 in the revised manuscript.

 L55: “high temperature performance”: it is very vague. Please, be more specific

Response: “high temperature performance” has been changed to “high electrical conductivities and high Seebeck coefficients”.

 L56: ”would be applied“: the message of this sentence is not clear.

Response: The sentence with “would be applied” has been changed to “The Cu-Se alloy core fibers with mechanical flexibility can be applied in flexible photoelectric detection, optical switch, thermoelectric sensing, and even multifunctional fiber sensing”.

 Figure 1: explain the colour code. (b-e) Element distribution of Cu, Se, O, and Si in the fiber core region. -> Element distribution of (b) Cu, (c) Se, (d) O, and (e) Si in the fiber core region.

Response: The color code means the high-low content and Figure 1 has been changed as below.

Figure 1. (a) Cross-section electronic micrograph of the Cu-Se core fiber; Element distribution of (b) Cu, (c) Se, (d) Si, and (e) O element in the fiber core region.

L83: what is “wave--- spectrometer”? L86: What is WDS and EPMA? If these are devices, please, rephrase the sentence to make it correct. Right now it is non-sense.

Response: “wave--- spectrometer” has been changed to “wavelength dispersive spectrometer” and “EPMA” has been changed to “electron probe micro analyzer”.

L92: Expand XRD. Figure 3: labels (a) and (b) are missing

Response: “XRD” in the label of Figure 2 has been expanded to “X-ray diffraction”. The labels (a)&(b) and the schematic of the electrical circuit of Figure 3 have been added as below.

Figure 3. (a) Absorbance spectra for CuSe raw powder and Cu-Se core fiber; (b) The current-voltage curves of Cu-Se core fiber in dark and irradiated conditions. Inset of (b) is the optical graph of a Cu-Se fiber and schematic of the electrical circuit.

Reviewer 2 Report

The authors have developed Cu-Se alloy core fibers with glass cladding by a thermal drawing method of a reactive molten core. The composition, crystallography, and photoelectric/thermoelectric performance of the fiber cores have been investigated. The manuscript is interesting and useful for the design and application of fibers. The paper is acceptable to be published in Nanomaterials, provided the following issue can be addressed

  1. Some abbreviations should be clarified when they appear for the first time.
  2. Add more details in the table to describe the physical parameters of the considered experimental setup.
  3. A figure is required to describe the experimental test.
  4. Add some discussion regarding the difference of this work and the authors’ preious work [15].
  5. Add some discussion either in the discussion Section or the Introduction regarding the potential application of the proposed fiber in optical communications and sensings.

See e.g.

C Jin et al., Nonlinear coherent optical systems in the presence of equalization enhanced phase noise, Journal of Lightwave Technology, 2021.

L Wang et al., CuO-modified PtSe2 monolayer as a promising sensing candidate toward C2H2 and C2H4 in oil-immersed transformers: a density unctional theory study, ACS omega, 2022.

Author Response

 Responses to the Reviewers’ comments:

 Reviewer #2:

The authors have developed Cu-Se alloy core fibers with glass cladding by a thermal drawing method of a reactive molten core. The composition, crystallography, and photoelectric/thermoelectric performance of the fiber cores have been investigated. The manuscript is interesting and useful for the design and application of fibers. The paper is acceptable to be published in Nanomaterials, provided the following issue can be addressed

Response: We thank the reviewer for the recommendation to publish this paper in Nanomaterials. The related responses to the issues are provided as follows.

 Some abbreviations should be clarified when they appear for the first time.

Add more details in the table to describe the physical parameters of the considered experimental setup. A figure is required to describe the experimental test.

Response: XRD is expanded to X-ray diffraction, EDS is expanded to wavelength dispersive spectrometer, and EPMA is expanded to electron probe micro analyzer. Revised Figure 3 and additional Figure 4 are shown as followings to the considered experimental setup and test. And the related revision has been made clear at Line 78-81 of Page 2 and at Line 146-151 of Page 5 in the revised manuscript.

Figure 3. (a) Absorbance spectra for CuSe raw powders and Cu-Se core fibers; (b) The current-voltage curves of a Cu-Se core fiber in dark and irradiated conditions. Inset of (b) is the optical graph of a Cu-Se fiber and schematic of the electrical circuit.

Figure 4. (a) Schematic of an electrical circuit connecting the two ends of a fiber or a micro-nano taper; Inset of (a) is the optical micrograph of a Cu-Se taper. (b) The thermoelectric voltages at temperature differences of 0 K, 3 K, 5 K, 8 K, or 10 K cross the two ends. The error bar in (b) shows 5% measurement uncertainty.

 Add some discussion regarding the difference of this work and the authors’ preious work [15].

Add some discussion either in the discussion Section or the Introduction regarding the potential application of the proposed fiber in optical communications and sensings.

See e.g.

C Jin et al., Nonlinear coherent optical systems in the presence of equalization enhanced phase noise, Journal of Lightwave Technology, 2021.

L Wang et al., CuO-modified PtSe2 monolayer as a promising sensing candidate toward C2H2 and C2H4 in oil-immersed transformers: a density functional theory study, ACS omega, 2022.

Response: Thanks to the reviewer for the constructive suggestions. Our previous study of [Advanced Materials, 2022, 34, 2202942] is mainly focused on the thermoelectric and mechanical performance of micro-nano Bi2Te3 fibers without any optical irradiation. And the related discussions have been added at Line 130-132 of Page 4 in the revised manuscript. The Cu-Se alloy core fibers indeed have potential applications for gas sensors. And the related discussions have been added at Line 27-28 of Page 1 and at References [2-7] in the revised manuscript.

Round 2

Reviewer 1 Report

Comments on:

  Multifunctional Cu-Se alloy core fibers

by Min Sun, Yu Liu, Dongdan Chen and Qi Qian

 The paper quality was significantly improved in the second revision. I have no objections to publish this study in “Nanomaterials”.

 Only one minor request from my side on Figure 1: please, explain the colour palette in the z-axis, i.e.  what values is denoted by the colour and how much is red/blues

Author Response

Responses to the Reviewers’ comments: Reviewer #1: The paper quality was significantly improved in the second revision. I have no objections to publish this study in “Nanomaterials”. Response: We sincerely thank the reviewer for the comment about publishing this study in “Nanomaterials”. And the related responses to the comments are attached below. Only one minor request from my side on Figure 1: please, explain the colour palette in the z-axis, i.e. what values is denoted by the colour and how much is red/blues Response: As the revised Figure 1 below, the color palette at right exhibits the relative concentration of the related element in (b-e) from low to high. The blue~red color has been normalized to 1~100. The related revision has been added to the label of the revised Figure 1. Figure 1. (a) Cross-section electronic micrograph of the Cu-Se core fiber; Element distribution of (b) Cu, (c) Se, (d) Si, and (e) O element in the fiber core region. The color palette at right exhibits the relative concentration of the related element in (b-e) from low to high.